# Safety in Public Open Green Spaces in Fortaleza, Brazil: A Data Analysis

**Bárbara Mylena Delgado da Silva [1,]*** , **Eszter Karlócainé Bakay [1]** and **Mariana Batista de Morais [2]**

1   Department of Garden and Open Space Design, Institute of Landscape Architecture, Urban Planning and Garden Art, Hungarian University of Agriculture and Life Sciences, 1118 Budapest, Hungary; karlocaine.bakay.eszter@uni-mate.hu
2   Department of Civil Engineering, Doctoral School of Earth Sciences, University of Debrecen, 4032 Debrecen, Hungary; engmarianamorais@gmail.com
*   Correspondence: barbaramdarquitetura@gmail.com; Tel.:+36-70-433-8568

**Abstract:** Latin America is as heterogeneous as its cities. To understand Latin American cities, it is necessary to have a clear vision of how they are organized, not only physically but according to their social, cultural, and economic contexts (which are associated). Historically, it has suffered a lot in terms of politics and the security of its cities. Insecurity reflects a structural problem; economic and social inequality are the main actors of spatial segregation, motivating violence and, consequently, the insecurity of urban space. Fortaleza is one of the largest Brazilian cities, and it is possible to fit it into this reality. Many public actions may benefit only one sector of society, showing biased investments and, again, confirming the tremendous economic and social differences in Latin American cities. In this study, questionnaires related to attendance, feelings, maintenance, and safety were made to some of Fortaleza's residents regarding an urban park called Parque do Cocó, one of the biggest in Latin America. Due to its large area, it is located in different city neighborhoods, allowing for us to see the differences in treatments throughout its extension. This article aims to understand how the public opinions and mentality of a portion of the population are characterized concerning safety in green public spaces in the city. In addition, the insecurity of public green spaces can also be inserted into a problem of environmental injustice in the urban context. This study of Fortaleza's Cocó Park highlights significant disparities in safety perceptions and maintenance across socioeconomic regions. Findings indicate that areas with higher human development index (HDI) scores experience better park conditions. The research underscores the necessity for comprehensive urban policies that address socioeconomic inequalities, as evidenced by the correlation between crime rates and HDI. Cocó Park emerges as a key factor in sustainable urban development, aligning with Fortaleza's urban planning goals. The study emphasizes the critical role of urban green spaces in enhancing the quality of life and fostering social cohesion in urban landscapes. Moreover, with the data collected, it will be possible to stress further how urban adequacy relates to social situations in Latin American cities.

**Keywords:** environmental injustice; urban parks; social-economic inequality

## 1. Introduction

This article explores the impact of socioeconomic inequalities on urban insecurity, focusing on urban green spaces, their qualities, and how accessible they are to the population. In Brazil, Fortaleza and Parque do Cocó serve as a case study to understand this dynamic. We aim to investigate how socioeconomic disparities influence the state of urban parks, which are key to cities' environmental and social health.

There is existing research on inequality and urban insecurity, but less attention has been given to how these issues manifest in urban parks and Latin American cities. These spaces are vital for environmental health and community well-being. In cities such as Fortaleza, where socioeconomic differences are pronounced, these disparities can lead to the poor maintenance and neglect of urban parks, affecting their sustainability and safety.

The literature review will examine the correlation between socioeconomic status, the condition of urban parks, and their role in promoting sustainable and safe urban environments. The findings of this study are expected to contribute to a deeper understanding of urban environmental injustices in Brazilian cities and urban parks. The research highlights that such injustices are equally prevalent in the daily life of larger cities. In Brazil, environmental injustice is a central concern in urban settings, affecting city dwellers' quality of life and security. This knowledge is crucial for developing urban planning and policies that aim to create more equitable, sustainable, and secure urban spaces for all residents.

The security issue in public open green spaces, such as urban parks, has become increasingly important in the development of modern cities. The availability of safe and secure public spaces is critical for the well-being of urban residents and the future growth of cities. According to the United Nations' 11th Sustainable Development Goal, which aims to "Make cities inclusive, safe, resilient, and sustainable" [1], ensuring safe public spaces is a fundamental aspect of sustainable urban development. As the World Health Organization emphasizes, quality of life refers to how an individual perceives their place in society within the framework of their culture and value systems and concerning their objectives, expectations, standards, and concerns, including worries such as violence and insecurity [2].

In particular, this issue is of great importance in developing countries, where these spaces are often limited in their total usage by the people due to the problem of insecurity. This academic article focuses on the issue of security in Latin American urban parks, specifically analyzing data from questionnaires about Cocó Park in Fortaleza, Brazil. The paper aims to shed light on the current security state in Latin America, specifically in a Brazilian urban park, through a data-driven approach. It also provides insights into the potential of these public spaces' safety and security while addressing the underlying social and economic factors contributing to insecurity.

Public open green spaces, such as urban parks, offer numerous benefits to cities and their inhabitants. Ricard T.T. Forman's research in the book Why Cities Need Large Urban Parks [3] highlights the critical role of parks in reducing floods, providing recreational opportunities, cooling the air, supporting biodiversity, and cleaning urban air. Furthermore, a dynamic balance between different park morphologies is essential for optimal park utilization, which can be achieved by incorporating large and small parks within a city. However, despite these benefits, public green spaces also pose several challenges, for example, the security topic [3].

Jacobs [4] commented on the vulnerability a park could provide to its surroundings in her book The Death and Life of Great American Cities. "Large parks are especially vulnerable to being 'dispirited border vacuums'. A single massive use of territory that produces danger and possible stagnation in surrounding urban neighborhoods" [4]. She even describes the border areas as active, contrary to people's imagination, and as just limits, which are essential in this dialogue between the city and the park. Although public parks are meant to be safe and enjoyable places for people to gather and participate in leisure activities, as previously commented, there are morphological aspects that can generate problems, including security problems.

Public open green spaces benefit communities, including recreation, relaxation, and social interaction opportunities. However, ensuring safety in these spaces can be a significant challenge. With the increased use of these areas, concerns about crime, accidents, and other incidents can arise, making it crucial to implement adequate safety measures.

Hence, what is considered a safe urban park? The guide from the U.S. Department of Justice [5] can describe it:

> "A 'safe' urban park is defined here as follows: A dynamic place where the design, maintenance, and policing of the park work together so that the general public perceives the park as a safe place, wants to go to the park regularly and spends their optional time in the park engaged in valued activities. Crime and disorder are limited, and diverse usage of the park by different groups is tolerated. Legal

activities are the dominant activities in the park. Because the local community values the park, it has a sense of "ownership" of it, and there is sufficient number of users who act as "natural guardians" to ensure informal social control". [5]

Through this description, we can base some essential words for a safe park. The design and diversity in use can influence people to use the space or not. The design must complement what the neighborhood needs and, for example, reflect the desires of a white and wealthy society. A neighborhood that does not have these users may react in a way that means the space is not used [6]. An unused space can also become unsafe. The "Natural Guardians" effect is another key that enables space security. According to Gehl [7] and Jacobs [4], the presence of people in public spaces indicates that the same space is good and safe. In her influential work "The Death and Life of Great American Cities", Jane Jacobs introduced the concept of "eyes on the street", fundamentally altering urban studies' approach to city planning. This concept emphasized the crucial role of community engagement in creating vibrant, safe urban spaces, challenging conventional urban renewal methods that overlooked resident experiences [4].

Other essential points to be commented on are the maintenance and policing of these spaces. Regarding the maintenance of public spaces, we must also consider that their quality can demonstrate that the region suffers from problems of social and economic inequality, which is also inserted in the context of environmental injustice and situations linked to a lack of security. "(...) Poor neighborhoods have more environmental problems than affluent neighborhoods, and that these include a greater range of problems, and problems that are more severe, particularly graffiti, litter, fly-tipping, and generally the poor maintenance" [8].

While vigilance can be a great tool against crime in public spaces, it can also create more fear among people who use it. "While policing, surveillance, and strict use regulations might increase the perception of safety, they can also contribute to accentuating fear by increasing distrust among users" [9].

When considering social theories about public space security, such as Jane Jacobs' "Eyes on the Street", we can emphasize that a greater variety of uses in these areas broadens the potential for public engagement. This, in turn, leads to a more natural form of surveillance by the community. Additionally, the "Broken Window Theory" underscores the importance of the physical maintenance of these spaces. According to this theory, if a broken window is left unrepaired, people will assume that no one cares about it and that there is no oversight. This perception can lead to more vandalism and a decline in the social order, conveying that societal norms and authority are not upheld. "One unrepaired broken window is a signal that no one cares, and so breaking more windows costs nothing" [10].

Security is a critical dimension to consider when striving to foster sustainable communities. Sustainability as a concept encompasses three fundamental pillars: the social, environmental, and economic. Notably, the issue of security intersects with all three of these pillars. This implies that environments lacking security can have a direct and adverse impact on people's quality of life, potentially resulting from social and economic inequality—a phenomenon inextricably tied to the social pillar. Furthermore, as articulated by Túlio Kahn [11], insecurity can also influence economic aspects, particularly concerning space usage and the fear of utilizing spaces at different times.

Lastly, from an environmental perspective, apprehension about occupying these spaces leads to their underutilization, perpetuating insecurity, as many authors, including Jane Jacobs and Jan Gehl, have theorized. When green spaces are marginalized due to societal fears, they not only lose their intended social function but also risk becoming spaces with diminished sentimental value over time, aligning with the "Broken Window" theory [10], which could lead to acts of vandalism and degradation. This apprehension poses a significant challenge, as these green areas can deteriorate over time despite their crucial roles in urban life as biological, psychological, and physical functions.

Security is a broad topic that is distinguishable at different intensity levels worldwide, and many aspects determine whether a space is safe. In addition to the distinct sense of

security, showing a sample percentage of crimes committed in one region where this group exists and what they feel and identify as insecure is different.

Túlio Kahn [11], in his article, "The Perception of Public Security in Brazil", in the 2017 victimization study carried out by the Brazilian Ministry of Justice and Public Security, comments that excessive insecurity is harmful because companies fail to settle in certain places, tourists are scared away, people stop going out at night to study or have fun, property values depreciate, and cities become vertical, causing people to lose the quality of life in their daily lives.

In order to talk more about the issue of security in Latin American cities, it is necessary to understand which factors are attributed to this reality. Many studies prove that social and economic differences in a location cause insecurity. "The segmentation of the city and social classes affects public spaces since not all people have easy access to them for recreation, either due to distances and mobility difficulties, as well as insecurity, lack of spaces, or neglect by authorities or centralism. Therefore, the phenomenon of the appropriation of public spaces could stem from some of these issues due to the need to access them in an easy, nearby, and/or affordable way" [12].

It is possible to cite the works of Juán Ruíz that comment on the violence in Latin American cities [13]. He expresses that violence is not homogenous and that the main reason for this violence is the product of economic and social inequality, which entails socio-spatial segregation. "Structural violence is not only the violent physical act but also the violence embedded in social structures characterized by high levels of inequality and that is exercised in a precise and systematic way on individuals and communities" [13].

Another author commenting on insecurity connected to economic and social inequality is José Manuel Pires Leal, who said: "There is no feeling of insecurity. What inhabits individuals is a plurality of ways of expressing fears built from the intersection of variables such as the dominant type of solidarity in the community or place of residence; the residential proximity to places marked by exclusion and socio-cultural and economic asymmetries; and the experience of direct or emotionally close victimization" [14].

In addition, Jan Gehl, in his book *Cities for People*, says:

> "Social and economic inequality is the backdrop for high crime rates and the fully or semiprivate attempts to protect property and private life. (. . .) It is important to point out that simple individual urban crime-prevention solutions are not of much help, where the invasive sense of insecurity is often deeply rooted in social conditions". [7]

The 2022 World Inequality Report shows various comparative analyses, graphs, and maps worldwide. Latin America has a situation of great inequality compared to European countries, and for example, in Brazil, the bottom 50% of the population earns 29 times less than the top 10%, "Wealth inequality in Brazil is also among the highest in the world. In 2021, the bottom 50% in the country owned less than 1% of total national wealth, whereas the top 1% of the population owns about half of total wealth" [15].

It is also necessary to point out that, although Latin American cities have insecurity as a point of comparison, we must remember that each one has its unique economic, cultural, and social context. In addition, people's tolerance regarding security can be reflected in different ways. Jan Gehl [7] says that parts of the world where cities have cultural traditions, family networks, and social structures are crucial for keeping crime low despite economic inequalities. "The sense of security is affected not only by personal experience with crime but by numerous other variables such as place of residence, age, gender, exposure and degree of trust in the media, among others" [7].

In the Report on Regional Human Development of 2021 by the United Nations Development Programme (UNDP), it is demonstrated through graphs that, in the Latin America and Caribbean (LAC) region, despite comprising only 9% of the global population, they account for one-third of global homicides, and these countries exhibit higher homicide rates than countries with similar levels of inequality [16].

The pandemic caused by COVID-19 was a factor that determined a drop in crime levels in Latin America. However, according to studies carried out by the LAPOP's 2021 Americas Barometer [17], even so, the feeling of security still shows that at least two in five say they feel somewhat or very unsafe. Haiti is at the top of the list, with 66% of people considering their neighborhood to be somewhat or very unsafe; Brazil is the 15th on the list with 42% [17].

According to the Brazilian public security yearbook [18], property crimes, particularly robberies, have a decisive impact on the population's sense of security. The yearbook presents evidence indicating a reduction in property crimes from 2019 to 2020, coinciding with the onset of the SARS-CoV-2 pandemic. However, recent statistics for the year 2021 suggest a considerable resurgence of property crimes, particularly in the wake of the increasing vaccination rates.

In Brazil, the perception of security is achieved through the absence or control of threats and illicit activities, significantly influencing the quality of life of its people. Public security is regarded as a duty of the state, yet it also represents a shared responsibility among all citizens. Insecurity is often linked to factors such as violence, social inequality, corruption, and social vulnerability, contributing to the escalation of urban violence and social tension. Urban violence is understood in terms of both crimes against life and property as well as through the perception and feelings of insecurity among residents of peripheral areas [19,20].

The inequality and insecurity described by Juán Ruíz [13], Jan Gehl [7], and José Manuel Pires Leal [14] also extend to public urban parks. These influences are also where we can find socio-space segregation and environmental injustice, and gentrification issues can arise. These green open spaces should be safe and inclusive for all. Nevertheless, when specific individuals and communities are excluded due to insecurity, they can become exclusive spaces dominated by only a select few.

This exclusion often stems from economic and social inequality, spatial segregation resulting from gentrification processes, and the effects of structural violence. As a result, many individuals and communities are unfairly denied access to the physical, mental, and social benefits that public urban parks can provide, perpetuating the cycle of environmental injustice [21]. "Redressing park-poverty in communities of color and/or low-income households can, however, create an urban green space paradox. As more green space comes online, it can improve attractiveness and public health, making neighborhoods more desirable. In turn, housing costs can rise. Such housing cost escalation can potentially lead to gentrification" [22].

Based on the 2022 Brazilian Public Security Yearbook [18] produced by the Brazilian Forum on Public Security, Brazil has a population that represents 2.7% of the world's population and, in 2020, accounted for 20.5% of known homicides committed worldwide. Apart from Brazil, only India and Mexico have a volume of records as extensive [18].

From the data provided in the Yearbook, it is possible to infer that Fortaleza is one of the most violent capitals among the Federative Units capitals in Brazil, as you can see in Figure 1, considering the obtained data on intentional violent deaths (IVD). The IVD in the yearbook is characterized as the sum of victims of intentional homicide (including femicides), robbery followed by homicide, bodily injury followed by death, and deaths resulting from police interventions both in and out of service. Some deaths resulting from police interventions are included in the tally of intentional homicides [18].

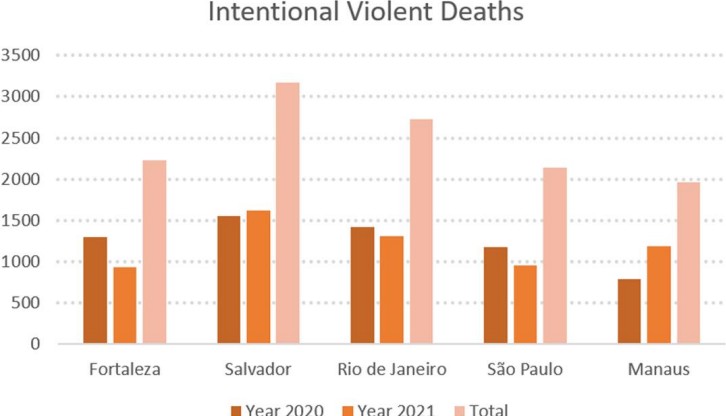

**Figure 1.** The graphic made by the author shows the 5 Brazilian capitals with the highest number of intentional violent deaths. Data from: State Secretariats for Public Security and Social Defense; Institute of Geography and Statistics (IBGE) [18,23].

Given the previous information, it is now necessary to provide a brief explanation of the administrative zoning of the city of Fortaleza as well as some primary data. According to data from the latest census conducted in 2010 by the Brazilian Institute of Geography and Statistics [23], Fortaleza ranks fifth among the largest populations in Brazil. It is the capital of its province (state), Ceará, with 2,703,391 inhabitants. Additionally, it has a population density of 7786.44 inhabitants per square kilometer [23].

According to the Secretariat of Public Security and Social Defense of the State of Ceará (CE), Fortaleza is divided into administrative areas called integrated security areas (AIS, from its acronym in Portuguese). These regions represent an amalgamation of multiple districts, constituting a distinct conjunction from the administrative regions, embodying another distinct limitation of geographical areas.

The integrated security areas comprise the public security administrative units of the city, and their division is shown in Figure 2. They are managed through shared governance between the Secretariat of Public Security and Social Defense (SSPDS-CE) units. This administrative division collects and analyzes data related to non-compliance with the law. Using this zoning makes it possible to conduct an analysis of these crimes. Subsequently, the following data collections pertain to theft incidents within the city, and another dataset relates explicitly to instances of lethal violent crimes, with geographic information presented in the form of maps, as seen in Figure 3.

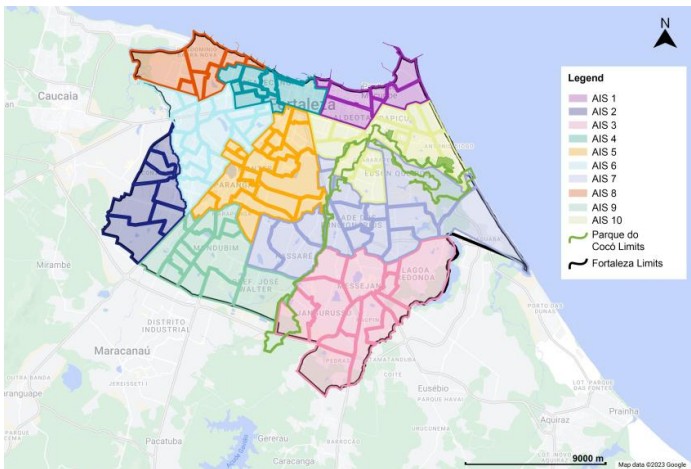

**Figure 2.** Map of integrated security areas (AIS) in Fortaleza made by the author with the data from the Secretariat of Public Security and Social Defense [24].

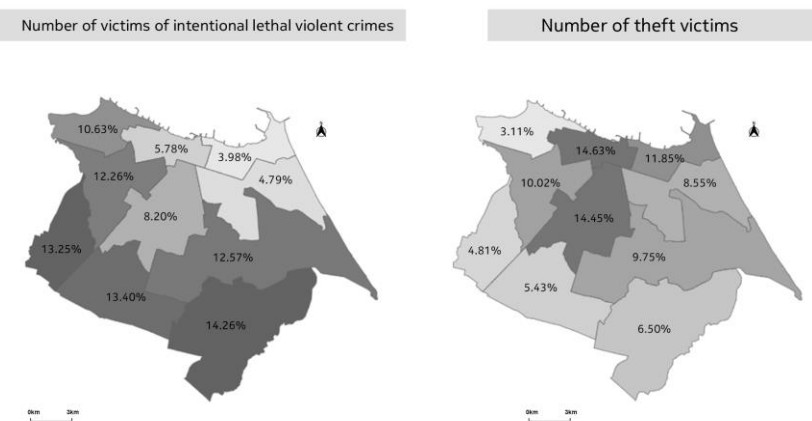

**Figure 3.** Maps made by the author with the data on thefts on the left and intentional lethal violent crimes (CVLI) on the right. Percentage data from the Secretariat of Public Security and Social Defense [23].

The Parque do Cocó in Fortaleza is the largest urban park in Brazil's northeastern region, situated along the lower Cocó River. This linear public park spans 1,155 hectares within Fortaleza's Metropolitan Region. The park's eastern sector is witnessing rapid urban and economic growth, becoming increasingly populated. This expansion is placing immense strain on the park's ecosystem, particularly the preservation area [25]. The Parque do Cocó encompasses the security areas AIS 10, AIS 07, and AIS 09; serves as a boundary; and shares space with AIS 03.

Within the confines of Parque do Cocó, the most popular spots for leisure activities are urban parks such as Parque Adahil Barreto and Parque do Rio Cocó, located in the AIS10, more formal urbanized areas from the entire park. The neighborhoods around these formal urbanized park areas show a marked enhancement in their human development index (HDI). However, the reality is starkly different in other parts of Parque do Cocó. These areas are characterized by noticeable pollution from debris, poorly defined and maintained boundaries, and the presence of unstable housing that is at risk of flooding from the river [25]. We can even observe that these are areas among the most dangerous regarding lethal violent crimes, such as AIS 03, AIS 07, and AIS 09.

Drawing together the key points discussed, this article aims to deepen understanding of the socioeconomic dynamics of insecurity in Latin American cities, specifically focusing on urban parks. This was further elucidated through a literature review. Additionally, the goal is to identify environmental injustices and their relationship to safety in Fortaleza's largest park, facilitated by creating and analyzing maps and questionnaires. Through this, we hope to shed more light on these issues with the aspiration of making a difference and providing more information that could contribute to future solutions for these pervasive problems.

## 2. Methods

This study utilized a mixed-methods approach that combined qualitative and quantitative data analysis to explore the security situation in public spaces, focusing on open green spaces. The study began with a literature review of security in public spaces and the situation of security in Latin America. The review provided a comprehensive understanding of the current security state in public spaces in Latin America. Furthermore, some quantitative analyses of social media open sources were combined with a mapping system for further investigation towards activities on the object of study.

A quantitative analysis was conducted on a collection of Instagram photographs tagged with "Parque do Cocó" as their location. This initial dataset consisted of the first 150 images retrieved using the search terms "Parque do Cocó" in the year 2021. Photographs not depicting the activities of the subjects or those evidently captured outside the park

boundaries, as well as non-photographic content (such as text-only updates or commercial advertisements), and images with indiscernible locations were excluded from the analysis.

The collection of images for creating an analytical map was conducted by searching Instagram for "Parque do Cocó". The assemblage included the first 150 images sourced using the search term "Parque do Cocó" in May of 2022. For the purpose of analysis, images that did not show the actions of the photographers were not considered. In the same way, images that obviously were not captured inside the park or were not actual photos (such as text-only posts or promotional ads) or those where the location could not be clearly identified were likewise omitted. Afterward, a map detailing the sites where the photos were captured was developed using visual identification of the areas, supplemented by personal visits to these locations and employing the street view feature of Google Maps. This enabled the quantification and charting of the exact locations of these photographs and the activities depicted in them.

Following this, a geographical mapping of the photo locations was established through visual identification of the sites. This process was enhanced by on-site visits and the utilization of Google Maps' street view feature, facilitating the quantification and geographical plotting of the photographic subjects and their activities.

For data collection, the study employed a questionnaire-based survey focused on an urban public park in Fortaleza. The questionnaire was designed to gather information about the security situation in the park, including maintenance and quality issues as well as participants' personal views of the space. The survey was distributed online to the inhabitants of the city. This survey was implemented via Google Forms and was disseminated organically through social media platforms and personal networks, accompanied by requests for further sharing. There was a total of 81 responses to the survey, with majority of respondents being residents living near the areas close to Cocó Park. The questionnaire conducted contained the following questions:

1.  Have you ever visited Cocó Park?
2.  Which neighborhood do you live in?
3.  How often do you visit Cocó Park?
4.  Do you consider Cocó Park a tourist attraction in our city?
5.  Which area(s) of Cocó Park have you visited or do you frequent, based on the image given? (Figure 4).
6.  If you do not frequent or have never been to Cocó Park, could you tell me why?
7.  If you do not frequent Cocó Park, which park in Fortaleza do you usually visit?
8.  If you have visited or frequent Cocó Park, what do you usually do there?
9.  How attractive do you find the park in terms of the activities offered?
10. How do you feel when you visit Cocó Park?
11. How safe do you feel there?
12. What about its state of maintenance?
13. Have you ever taken photos at Cocó Park?
14. If yes, what caught your attention and made you want to take those photos?
15. What would encourage you to visit Cocó Park more often?

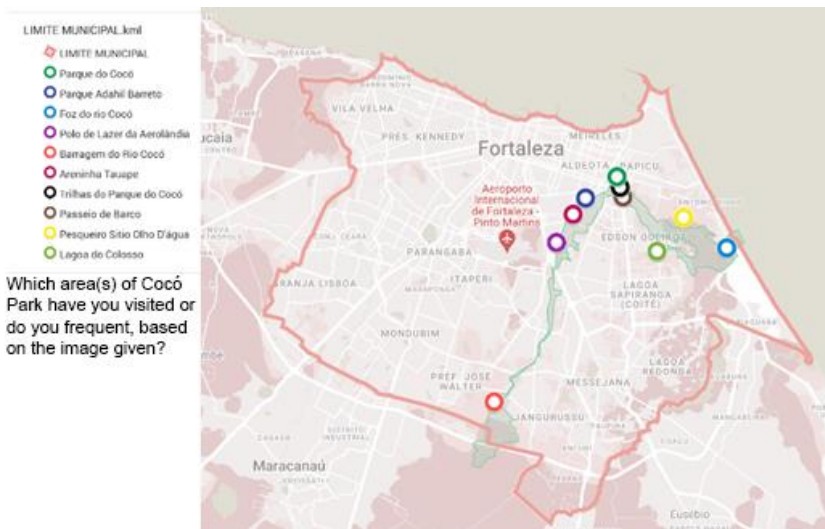

**Figure 4.** Map made by the author, with the more relevant locations in Parque do Cocó from Google Maps.

The fusion of social media analysis, geographical mapping, and public survey in this study provided a holistic view of the park's security situation, reflecting the complexities and dynamics of public space usage in Latin America. This methodological blend not only shed light on the current state but also set a precedent for future studies in similar urban settings.

## 3. Results

### 3.1. Mapping Inequality

With the forthcoming information, the creation of maps was facilitated, thereby simplifying the visual identification of overlapped data.

By integrating data on the neighborhood human development index (HDI), the percentage of violent crimes, and the geographical location of Parque do Cocó, it was possible to generate the map in Figure 5.

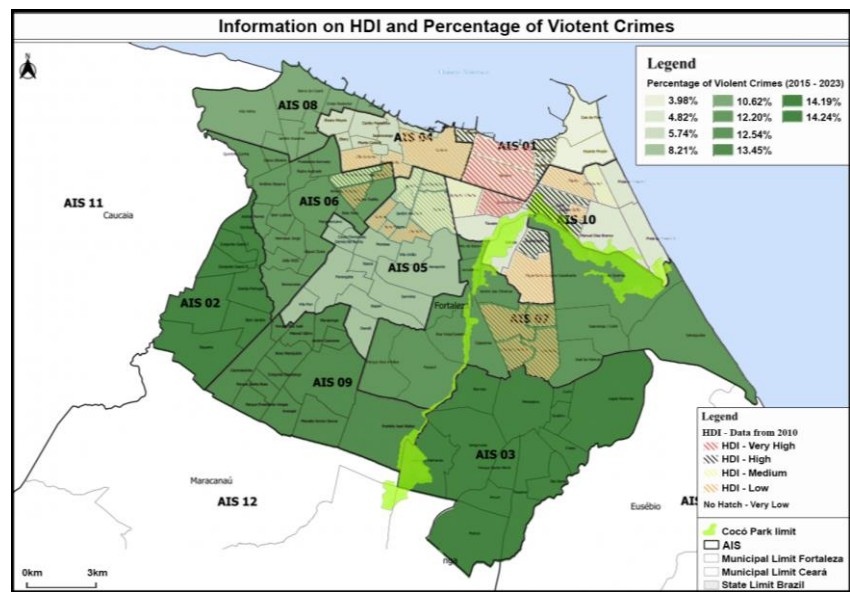

**Figure 5.** Map made by the author, with the base map from SUPESP-CE (2000) showing the percentage of CLVI in the different AIS in Fortaleza, the Paque do Cocó region, and HDI.

In a local newspaper article titled "Crime Geography: Fortaleza Areas with the Lowest HDIs Concentrate the Highest Homicide Rates in 2020" [26], it is reported that the neighborhoods comprising the capital's integrated security areas 2, 3, and 9 exhibit the lowest human development index values and the highest records of lethal violent crimes. Therefore, based on the created map and the article itself, we infer that the most violence-affected areas in the city are the fringe regions.

Further analyzing the map, it is evident that most of the Parque do Cocó area is situated within regions exhibiting a higher incidence of lethal violent crimes, encompassing numerous neighborhoods with notably low HDI scores. Exceptions to this pattern are limited to small areas where a cluster of neighborhoods displays a significantly better HDI and a lower percentage of lethal violent crimes.

Through the analysis of the map above and in conjunction with another study conducted on environmental justice in Fortaleza and Goiânia [25], it is possible to discern that the southern zone of the municipality and the boundaries of Cocó Park are where the highest rates of violent crimes are concentrated, along with the largest zones of precarious settlements in risk areas [25]. This may indicate the disparity in access to high-quality green spaces for the population of the city of Fortaleza, thereby demonstrating an environmental injustice along the park. "The park comes to represent in itself a form of urban planning, which disposes individuals unequally, but which incorporates a process of daily insurgency, of resistance to forms of imposition and standardization of practices" [27].

Over the course of several years of living in the city and closely monitoring the urban infrastructure and public safety development within this specific region, it becomes apparent that this more developed area has become increasingly popular among the population. However, this observation is not solely based on qualitative assessments but is also supported by the quantitative analysis of Instagram photos, where Parque do Cocó was tagged as the location of the pictures.

The initial collection comprised the first 150 photos obtained through a search using the keywords "Parque do Cocó" in 2021. For analysis purposes, photos that did not depict the activities of the photographers were disqualified. Similarly, photos that were clearly not taken within the park were non-photographic posts (such as text-only posts or sales advertisements). Those in which the location was indistinguishable were also excluded; the table is shown in Figure 6.

| Parque do Cocó | | | | | | | | | | | | | | | | |
|---|---|---|---|---|---|---|---|---|---|---|---|---|---|---|---|---|
| Number of Photos disregarded | | | | | Activities — Number of Photos Considered | | | | | | | | | | | |
| Photos only of Wild Animals and Plants or the water | Inside the Apartment | Fire | Drone Photos | Publications without photos; not distinguishable or not applicable | Trail walks | Off-trail walking or running | Picnic | Walk the dog | Off-trail walks | Professional photography | Religious meeting | Government or ONGs activities | Skate | Cycling | Boat tour | Total |
| 0 | 0 | 0 | 1 | 3 | 6 | 0 | 2 | 0 | 3 | 0 | 0 | 0 | 0 | 0 | 0 | 15 |
| 1 | 2 | 2 | 1 | 0 | 3 | 3 | 1 | 0 | 1 | 1 | 0 | 0 | 0 | 0 | 0 | 15 |
| 1 | 1 | 1 | 0 | 0 | 2 | 0 | 2 | 0 | 5 | 0 | 1 | 1 | 0 | 0 | 1 | 15 |
| 0 | 1 | 0 | 0 | 2 | 6 | 1 | 1 | 1 | 1 | 1 | 0 | 0 | 1 | 0 | 0 | 15 |
| 1 | 0 | 0 | 0 | 1 | 3 | 0 | 0 | 0 | 8 | 1 | 0 | 0 | 0 | 1 | 0 | 15 |
| 1 | 1 | 0 | 1 | 3 | 3 | 0 | 0 | 0 | 3 | 1 | 0 | 0 | 0 | 1 | 1 | 15 |
| 3 | 0 | 0 | 1 | 1 | 1 | 0 | 3 | 0 | 3 | 2 | 0 | 1 | 0 | 0 | 0 | 15 |
| 0 | 2 | 0 | 0 | 1 | 4 | 2 | 0 | 0 | 5 | 0 | 0 | 0 | 0 | 0 | 1 | 15 |
| 2 | 1 | 0 | 0 | 1 | 3 | 1 | 0 | 1 | 6 | 0 | 0 | 0 | 0 | 0 | 0 | 15 |
| 2 | 2 | 0 | 0 | 1 | 2 | 1 | 1 | 3 | 2 | 0 | 0 | 0 | 0 | 1 | 0 | 15 |
| 11 | 10 | 3 | 4 | 13 | 33 | 8 | 10 | 5 | 37 | 6 | 1 | 2 | 1 | 3 | 3 | 150 |
| 7% | 7% | 2% | 3% | 9% | 22% | 5% | 7% | 3% | 25% | 4% | 1% | 1% | 1% | 2% | 2% | 100% |
| 27% | | | | | 73% | | | | | | | | | | | 100% |

**Figure 6.** Table with the counting of images on Instagram and activities made by the author.

Subsequently, a map of the locations where the photos were taken, which is shown in Figure 7, was created through the visual recognition of the sites, aided by visiting the places and using the street view tool of Google Maps. This allowed for quantifying and mapping where these photos were taken and the activities being conducted.

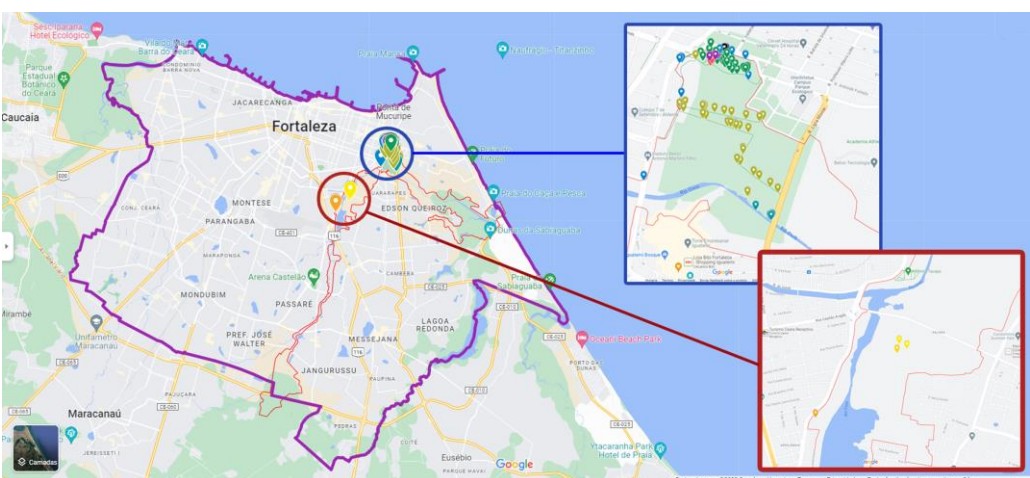

**Figure 7.** Map with recorded Instagram activities made by the author.

Property crimes, particularly robberies, profoundly influence the public's feeling of safety. A survey conducted in April 2022 evaluated the sentiments of people living in São Paulo and Rio de Janeiro on several issues. Concerning public security, the survey revealed that a striking 90% of participants harbored fears of street robberies [18].

Another online survey, conducted in July 2019 by the news website Mobile Time and the research solutions company Opinion Box, involved 2532 Brazilian internet users who own smartphones. The study revealed that 84% of Brazilian internet users with smartphones avoid answering calls on the streets, and 47% have experienced theft or robbery of their devices [28].

### 3.2. Mapping the Survey

The survey also provided percentage data on the demographics most targeted in robberies in Brazil, including gender, social class, age, and region. The incidence between men and women is nearly equal, with women at 46% and men at 48%. Age-wise, the most frequent targets are youths aged 16 to 29 at 52%, followed by adults aged 30 to 49 at 44%, and, finally, individuals 50 years or older at 34%. Regionally, the highest percentages of thefts are in the north at 65% and the northeast at 54%, followed by the central-west at 47%, the southeast at 45%, and the south at 34% [28].

From this data analysis, it becomes evident that the vast majority of Brazilians are apprehensive about using their smartphones in public spaces. Additionally, noting that the northeast region has one of the highest percentages of smartphone thefts in Brazil, we can infer that, based on the proportions of these photos, this specific area is more widely embraced and visited by the population as a safer public green space.

In addition to the aforementioned sampling method, data were also gathered through an online questionnaire. The questionnaire was conducted online using Google Forms. It was disseminated spontaneously through social media and close personal networks, accompanied by requests to share it with others further. This collection of data resulted in a map shown in Figure 8.

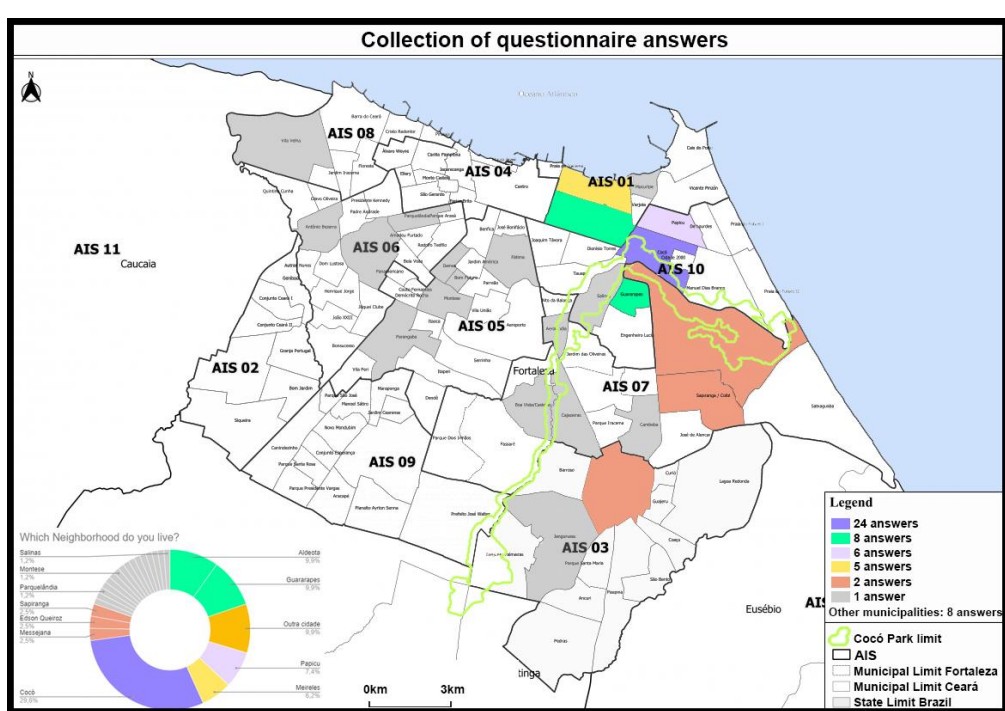

**Figure 8.** The map was made by the author with the base map from SUPESP-CE (2000), with the answers from the online questionnaire, and the location of the Parque do Cocó limits and integrated security areas (AIS) of Fortaleza.

There was a total of 81 responses, and the preponderance of respondents primarily comprised residents living close to Parque do Cocó, in areas characterized by a higher human development index (HDI), specifically categorized as "very high" and "high". Such responses collectively account for 39.5% of the total survey data.

Of the respondents, from the amount of 81 answers, 84.4% confirmed having taken photographs at Cocó Park. Regarding the respondent's perception of safety, as assessed on a scale ranging from 1 to 5, where 1 is not safe and 5 is very safe, 35% indicated feeling more secure than the average, marking a rating of 4 on the scale.

In the questionnaire, respondents were also asked about factors that would encourage them to visit Parque do Cocó more frequently. Multiple-choice options were provided, and the results indicated that 46.7% of respondents ranked "enhanced security" as their top preference. This was followed by "more dining options" at 36% and subsequently by "improved trail maintenance" and "better upkeep of urban furniture" at 33% and 32%, respectively, although it is also interesting to note that the majority of people, 36.7%, believe that the park's maintenance condition falls within the range of 4 on a scale from 1 to 5.

The individuals who do not visit or have never been to Parque do Cocó were also asked about their reasons for not going. An amount of 42.3% of respondents mentioned "I don't have the time to go", followed by "I live far away, and it's challenging to go without a car" with 38.5% of responses and, in third place, "I consider it unsafe" with 19.2% of responses.

A map was also provided with regions known throughout the entire perimeter of Parque do Cocó, and people were asked which of these places they had visited or had ever been to. The top three responses were located near the previously mentioned areas, encompassing higher social and economic development regions. It is also important to note that the central area along the perimeter of the Parque Estadual do Cocó is heavily impacted by irregular occupations. It has a very narrow space, often comprising only narrow strips of vegetation and the river. Consequently, the population cannot recognize this space as a suitable recreation area.

## 4. Discussion

Through the collection and analysis of these data, it was possible to observe that the studied region is extensively used by people, not only from the nearby area but also as an attraction recognized by residents from many other neighborhoods and municipalities. However, it poses challenges for those who live far away and do not have private vehicles, making access difficult.

Additionally, despite the fact that the people who answered the survey consider the area safe, there is still concern about local security and the perception of park maintenance. In general, respondents believe the park is well maintained, but when specifically discussing urban furniture, improvements could be made. It is also noteworthy that an environmental injustice is discernible along the park's length, indicating the city's significant social and economic disparities. Consequently, this is also connected to the security conditions within the city.

In the survey, 96.3% of respondents confirmed that they had visited Cocó Park, with 88.9% considering it a significant tourist attraction within the city. These statistics underscore the park's potential for tourism and economic development. This also indicates that a portion of the population highly cherishes the park, which is perceived as significant tourist importance.

It can also be inferred from the photo analysis that people feel comfortable and safe using the space, where they feel at ease taking out their smartphones in public to take photos they want or of what they appreciate in the park. However, this reality applies only to a specific area of the park, which is located in a region with higher human development index (HDI) levels compared to the overall city of Fortaleza. The data show that the region with more favorable socioeconomic indicators could serve as a model for the broader park.

The maps revealed that the southern areas of the city experience a higher incidence of violent crimes, while the entire city, particularly the northern part, suffers from crimes such as robberies. By correlating this with the human development index (HDI) data of these regions, it can be inferred that areas most affected by robbery are those considered to have residents with better living conditions, especially the city center, which is the hub of commercial activity. Meanwhile, the southern region faces more significant structural challenges.

Since this survey was conducted by sending it to close contacts and further redistributed by them, it is evident that there is a concentration of responses from a specific community. This community does not represent the entirety of the city's population. However, it is noteworthy that these responses reflect a community in a better socioeconomic situation. This aspect cannot be disregarded; it demonstrates how this portion of Cocó Park effectively serves this community. Moreover, it shows that these respondents perceive the park positively regarding safety. It can also be considered that, since the region is more socioeconomically developed, the inequalities directly influence the use of the city's green spaces, as previously described in the literature review.

It is crucial to recognize that addressing the challenges related to the lack of maintenance, insufficient activities, and security issues in certain park areas is not solely a matter of physical renovation or restoration. It also necessitates governmental interventions to improve the prevailing social disparities within the neighborhoods surrounding the park.

In the city of Fortaleza, the paucity of public green spaces in the southern region exhibits a concerning correlation with social and security indicators. This area, already characterized by low human development index (HDI) scores, also contends with heightened rates of lethal violent crimes. The absence of green spaces, which are pivotal for fostering health, well-being, and social cohesion, may be intrinsically linked to these challenges.

An article by Joseli Macedo and Mônica A. Haddad, analyzing the equitable distribution of open spaces using spatial analysis to assess the urban parks of Curitiba [29], a Brazilian city in the south, also highlights the unequal access to parks. Their research found that the majority of parks are located in affluent neighborhoods. Even though people may travel by bus to use the parks, it is primarily those who live or work near them who utilize

these spaces. The study also indicates that neighborhoods in need of improved access to quality open spaces are predominantly situated in the southern part of the municipality, where there is a higher incidence of crime and a lack of recreational opportunities. Despite utilizing different analytical methods, the results are similar to those in this study.

## 5. Conclusions

The literature review enabled the identification of one of the main reasons for insecurity being socioeconomic inequality. It was found that environmental injustices and gentrification are intertwined with social situations. It is also crucial to recognize that these phenomena occur at different scales and that urban parks are among the spaces impacted by these social circumstances. According to the authors, there are theories that, if properly implemented, can contribute to the creation or maintenance of safer spaces. However, it is important to mention that this is just one of many factors to consider. As previously mentioned, the substantial socioeconomic disparities are much more complex and require significant governmental effort to improve the current situation in Latin American cities.

Green areas with proper care provide environmental benefits and serve as venues for leisure and social interaction, contributing to forming a more harmonious and resilient community. Their absence, therefore, may exacerbate already unfavorable living conditions, reflecting the diminished quality of life and potentially fueling cycles of violence and social inequality. This scenario underscores the critical need for integrated public policies that address urban development and socioeconomic issues, aiming to enhance the quality of life and reduce criminality in Fortaleza's southern region.

This highlights the importance of focusing on urban green spaces where society not only feels safe in using them but can also fully embrace the enhanced quality of life they have the potential to offer to the population. The Fortaleza government's urban development plans, including the 'Fortaleza 2040' plan [30], which integrates short-, medium-, and long-term sustainable development goals described by the United Nations [1], signify the city's commitment to sustainable urban development, but even with this commitment, the city is still far from achieving the highlighted objectives. Therefore, discussing Cocó Park is of great importance. It is necessary to comment on how crucial access to quality green spaces is and how it significantly influences society. Additionally, the aspects of violence and insecurity are also interconnected with this issue. It is evident that Cocó Park plays a multifaceted role within the city, encompassing biological, social, touristic, and physical aspects, thus serving as a prime candidate for the development of sustainable communities in the future.

With the study, it was possible to identify that the most violence-affected areas in Fortaleza are the peripheral regions. In addition, most of Parque do Cocó bordering regions on the southern zone present high rates of lethal violent crimes, low HDI scores, and large zones of precarious settlements in risk areas. In the questionnaire, respondents were asked about factors that would encourage them to visit Parque do Cocó more frequently, and 46.7% of respondents marked "enhanced security" as the top priority. Furthermore, the number of photos posted on Instagram in the same location of the park indicates that the area receives more visitors and can be perceived by the population as a safer spot.

Future practices and policies can be based on the idea of providing safe urban green spaces as an attempt to ensure more visitors, apart from the high quality and enough quantity of urban furniture and frequent maintenance, as pointed out by those who answered the questionnaire. Moreover, investing in urban green spaces, where the presence is sparse, can lead to a more just access of the population and enhanced life quality. In addition, investing in recovering "abandoned" public spaces, at times considered unsafe, ensures this one aspect of environmental justice in the city.

The number of questionnaires answered represents a convenience sample of the population. For future investigations, the survey can be extended to a broader public, and a more formal sampling can be used for a larger study. A larger collection of photos would be required for a more in-depth analysis, which could be a direction for future research. The

methodology used a sum of media analyses, geographical mapping, and public surveys, which can be applied in future studies for similar urban settings.

**Author Contributions:** Conceptualization, B.M.D.d.S., E.K.B. and M.B.d.M. methodology, B.M.D.d.S. and M.B.d.M.; validation, B.M.D.d.S. and E.K.B.; formal analysis, B.M.D.d.S.; investigation, B.M.D.d.S. and M.B.d.M.; resources, B.M.D.d.S.; data curation, B.M.D.d.S.; writing—original draft preparation, B.M.D.d.S.; writing—review and editing, B.M.D.d.S., E.K.B. and M.B.d.M.; visualization, B.M.D.d.S.; supervision, B.M.D.d.S. and E.K.B.; Project administration, B.M.D.d.S.; funding acquisition, B.M.D.d.S. All authors have read and agreed to the published version of the manuscript.

**Funding:** This research received no external funding.

**Institutional Review Board Statement:** No institutional approval was required to conduct the study.

**Informed Consent Statement:** Informed consent was obtained from all subjects involved in the study.

**Data Availability Statement:** Data are contained within the article.

**Conflicts of Interest:** The authors declare no conflicts of interest.

## Abbreviations

CLVI (acronym in Portuguese): Intentional Lethal Violent Crimes, AIS (acronym in Portuguese): Integrated Security Areas, HDI: Human Development Index. SUPESP-CE (acronym in Portuguese): Superintendency of Research and Public Security Strategy of the State of Ceará, IBGE (acronym in Portuguese): Brazilian Institute of Geography and Statistics.

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
