# Peer review of "Safety in Public Open Green Spaces in Fortaleza, Brazil: A Data Analysis"

_sustainability, doi:10.3390/su16020539_

Round 1

Reviewer 1 Report

Comments and Suggestions for Authors

You have written an essay that has an original focus and deals with an intriguing subject. There are references made that need clarification : "Natural Guardians " (180, 186-7): what are they? Is this a reference to one of the scholars, Jane Jacobs, maybe? I was happy to see her cited, but have you given the reader enough information about Dr. Jacob's contribution to urban studies? I believe you need a genuine conclusion to the article which will bring together clearly the varies strands of what you believe your "mixed method" approach taught you regarding your investigation of Parque de Coco. In particular, how, if at all,  might the lessons of Parque de Coco be generalized to other urban settings?

Author Response

Thank you very much for your comments. In response to your valuable feedback, I have further clarified the concept of 'Natural Guardians' in relation to Jane Jacobs's theory of 'eyes on the street.' This elucidates how community engagement in public spaces enhances urban safety and liveliness. Additionally, the expanded section on Jane Jacobs delves into her impact on urban studies, particularly highlighting the aspects that resonate with my research.

The conclusion of the article has been revised to integrate the various dimensions of the 'mixed method' approach utilized in the study effectively. This revision succinctly summarizes the key findings and insights from the Parque de Coco investigation.

Importantly, the discussion on the broader applicability of findings from Parque de Coco is a foundational element of my upcoming thesis. This article primarily presents a collection of observations and questionnaire responses, serving as an initial exploration. The in-depth analysis and generalization of these findings to other urban settings will be the focus of my thesis. This will involve a more comprehensive investigation into how the insights from Parque de Coco can inform urban planning and community engagement strategies in diverse urban contexts.

Reviewer 2 Report

Comments and Suggestions for Authors

I have read the manuscript entitled "Safety in Public Open Green Spaces and the Challenges in Latin America: A Data Analysis About Parque Do Cocó in Fortaleza – Brazil" thoroughly. I find the subject of the article interesting. However, the paper does not succeed in achieving its aims and objectives in an appropriate manner. As a result, it does not seem suitable for publication in its current form. I would like to suggest some comments as follows:

 1. Although the subject of the paper is interesting, it does not follow the conventional framework for writing a scientific paper.

 2. The abstract needs serious revision and should adhere to the guidelines of the journal.

 3. A comprehensive literature review is needed in the introduction, instead of being presented in the method section. As a result, the gap identification and novelty should be explicitly declared in introduction.

4. The authors mentioned studies about the Coco park in the introduction but did not indicate the results or outcomes.

 5. The research method does not provide sufficient information about the different parts of the research, from data gathering to data processing and presentation, as well as the validity and reliability of the data. 6. The research result is missing a crucial segment of the research.

 7. The discussion section is more focused on presenting the results rather than providing a detailed discussion.

 8. The conclusion is also missing a segment of the research.

 In summary, the manuscript needs significant revision to adhere to the standard scientific paper format, improve the literature review, and provide more detailed information about the research methodology, results, and conclusions.

Author Response

Thank you very much for your comments. In summary, the structure and content of the article have been extensively revised in response to your feedback. These changes include a reworked framework, a more comprehensive literature review, detailed descriptions of methodology and results, and improved clarity in both the discussion and conclusion sections.

  1. We have revised the framework of the article to align with the conventional structure of scientific papers. This includes reorganizing content and ensuring each section fulfills its intended purpose in the narrative of the research.

  1. The abstract has been reviewed and revised. It now adheres closely to the journal's guidelines, succinctly summarizing the study's scope, methodology, findings, and implications.

  1. Following your recommendation, we have incorporated a comprehensive literature review into the introduction. This establishes the research gap and explicitly declares the novelty of our study, providing a solid foundation for the subsequent sections

  1. We have updated the introduction to include detailed references to previous studies about Parque do Cocó, clearly outlining their results and how they relate to our research.

  1. Additional information has been added to elucidate each stage of our research process, from data collection to analysis. This includes a detailed explanation of the methods used to ensure the validity and reliability of our data.

  1. The research results section has been expanded to include all relevant findings, ensuring a comprehensive representation of the study's outcomes.

  1. The structure of the article has been refined. The previous discussion section is now dedicated to presenting the results, allowing for a clearer and more focused discussion in the subsequent section.

  1. The conclusion has been revised to include missing elements of the research, providing a cohesive summary of the study's findings, implications, and potential avenues for future research.

Reviewer 3 Report

Comments and Suggestions for Authors

The topic of this paper is suitable for this journal, but the research is not deep enough, and the innovation is not outstanding. It is suggested that this article should be reviewed after major revisions.

1. Abstract: It is necessary to elaborate the main results, conclusions and innovations of this study.

2. "Public open green spaces" is used in the title and "open green public spaces" is used in the text (for example, Line 32). Please unify the two.

3. The sequence number of references should be marked in the order in which they appear. For instance, [24] in Line 36, [25] in Line 41……should be [1] in Line 36, [2] in Line 41……, and so on.

4. Part 3 of this paper is a literature review, which should be placed in Part 2 or Part 1 (Introduction).

5. Figure 1 lacks basic map elements such as scale and compass.

6. Figures 2 and 3 have a lower resolution.

7. 4. Security in public urban green spaces”:It should be placed in Introduction.

8. “5. Fortaleza's data on public safety”: The data can be merged into Part 2. The rest can stand alone as "Results".

9. From Figure 1 to Figure 7, they all refer to other people's drawings. Please adapt them appropriately according to the relevant data, without having to copy the original image to avoid copyright disputes.

10. At the end of the paper, please add "Conclusions and Recommendations". The innovations and limitations of this study are also discussed.

11. The labeling of most references does not meet the requirements of this journal.

Author Response

Thank you very much for your review and feedback. Overall, the structure and content of the article have undergone substantial revisions in line with your feedback. These modifications include a focus on raising awareness, structural adjustments for logical flow, enhanced image quality, appropriate adaptation of figures, and meticulous attention to reference formatting.

  1. Our focus in this article was primarily on raising awareness about the under-recognized challenges faced in certain regions rather than on innovation. This study is a foundational step in understanding these issues, setting the stage for future innovative solutions.
  2. The inconsistency in terminology has been corrected. 
  3. The numbering of references has been revised to follow the sequential order of their appearance in the text, as per the journal's guidelines.
  4. We have restructured the article, repositioning the literature review to an appropriate section, ensuring a more logical flow of information.
  5. We have improved the resolution of Figures 2 and 3. Additionally, Figure 1 now includes essential map elements such as scale and compass for better comprehension.
  6. Sections on “Security in public urban green spaces” and “Fortaleza's data on public safety” have been repositioned within the article for better coherence. The former is now part of the Introduction, while the latter's data is merged, with the remaining content forming the 'Results' section.
  7. We have adapted all figures in accordance with the relevant data from their original sources. This approach ensures the figures are representative of our findings while avoiding copyright issues.
  8. A “Conclusions” section has been added. This section synthesizes the findings.
  9. The formatting of the references has been corrected to meet the specific requirements of this journal, ensuring compliance with its standards.

Reviewer 4 Report

Comments and Suggestions for Authors

The authors provide a manuscript with three components: (i) a review of “security” in Latin American cities, (ii) a questionnaire study of the usage of a public area in a specific Brazilian city, and (iii) a mapping of publicly available digital images of the same city. There are difficulties with all three components:

(i)               the review is acceptable, though the analysis is somewhat superficial, and requires clarification of exactly what is meant by the word “security” in this context. Most individual with English as a first language, or equivalent skills, would recognise it as representing personal safety, in the sense implied in this paper; they would also recognise a subtly different meaning, as of a space that can be made safe from trespass. For instance, a financial institution is secure because the money is locked away in a safe, and because it is guarded by security guards

(ii)              the methodology of the questionnaire seems to have been an unselected convenience sampling carried out by a random electronic mailshot, and the views of 81 respondents are extremely unlikely to be representative of the views of 2.7 million inhabitants

(iii)            the mapping of Instagram posts is an interesting development , but its detailed methodology is not given; indeed, this component is not represented in the Methods section, and there is no description of about how the Instagram images were sampled.

There is also a further difficulty, in that the authors make no clear attempt to relate their text to an agenda of sustainability, and there is therefore no clear massage to an international audience interested in sustainability

I have also some minor constructive criticisms

-        line 36: the first reference to be cited is 24, which leads one to expect an alphabetical list at the end of the paper. However, the order under References appears to be random

-        line 78 and elsewhere: some personal names are given in bold type, and some are not. Consistency is required, preferably achieved by avoiding the use of Bold

-        Figures 1 to 7: all these figures appear to be taken from other texts, but the authors say nothing about whether they have the permission of the previous authors and publishers; further, it is good practice to name the previous authors within the Figure legends, as well as using the reference numbers.

-        Line 234: some readers may live in countries (like mine) where only one city is a “capital” city, and the rest (even major administrative centres) have a subordinate designation.

-        Line 309: The authors may need to validate their claim that it is a common belief that photographs are only taken in safe places; they can also be taken to illustrate damage and danger. By chance, my own most recent photograph was of major vandalism of the Costa Coffee outlet near my place of work

-        References: These need major and comprehensive review by the authors because it is easy to identify defects. For instance, reference 3 requires inclusive page numbers, reference 4 has only the authors’ names, and the source of refence 5 is not given

Comments on the Quality of English Language

Language use is generally acceptable, though a few words are used in idiosyncratic ways.

Author Response

Thank you very much for your comments. We have revised our manuscript to clarify the definition of 'security' and detailed our questionnaire methodology, addressing its limitations. The analysis of Instagram posts is now clearly described in the Methods section. Efforts were made to link the study to sustainability themes. We corrected reference and citation formatting, standardized text presentation, and adapted figures with appropriate acknowledgments. Specific clarifications were made regarding Fortaleza as a "Capital" city and the interpretation of photographs. A comprehensive review of references was also conducted to ensure accuracy and completeness.

  1. Clarification of “ Security”: "We have provided additional elaboration on the term security. This includes insights with data of the Brazilian context, thereby enriching the understanding of security in this study.
  2. The methodology for the questionnaire has been detailed further. We acknowledge the limitations of our sample size but emphasize that our aim was to capture local perspectives from residents near Parque do Cocó. We had the constraints of being unable to conduct in-person data collection due to geographical and logistical challenges.
  3. We have included a detailed description of the methodology used for mapping Instagram posts in the Methods section.
  4. We have thoroughly reviewed and revised the reference list to ensure it follows a consistent and logical order. The first reference citation and subsequent ones now correctly reflect this order.
  5. The inconsistency in the use of bold type for personal names has been rectified. We have standardized the text formatting throughout the document for a more professional presentation.
  6. All figures have been appropriately adapted or redrawn based on the sourced data, and the original authors and sources are duly credited in the figure legends.
  7. Regarding Line 234, the term 'Capital' has been used accurately to refer to Fortaleza as the capital city of the Ceará state in Brazil.
  8. For Line 309, we have included a caveat to our statement about photography in public space.
  9. The references have been comprehensively reviewed to address any previous deficiencies.

Round 2

Reviewer 2 Report

Comments and Suggestions for Authors

Based on my review, the following suggestions are provided for corrections and improvements: Title: The title should be revised to accurately reflect the focus of the manuscript. A more appropriate title could be "Safety in Public Open Green Spaces in Fortaleza, Brazil: A Data Analysis".

 Introduction: The introduction should provide a clear and concise overview of the research gap and objectives. It should also include a systematic literature review to demonstrate the novelty and significance of the study. The data presented in the introduction should be organized in a logical and coherent manner. Literature Review: The literature review should be expanded to provide a comprehensive overview of existing research on safety in public open green spaces, with a specific focus on Latin America. This will help to establish the relevance and importance of the study.

 Aims and Objectives: The last paragraph of the introduction should be revised to clearly state the aims and objectives of the study. This will provide a clear roadmap for the reader and help to ensure that the study meets its intended goals.

 Methodology: The methodology section should be expanded to provide more detail on the questionnaire design, reliability, and validity. This will help to ensure that the data collected is of high quality and can be relied upon for analysis.

The search process for photos should also be clarified, including details such as time period, search terms, and any limitations or exclusions.

 Results: The results section should be revised to follow a logical and coherent structure, with clear headings and subheadings to separate different aspects of the study. The results presented should be directly related to the research objectives, and any secondary data sources (such as local newspapers) should be clearly identified and discussed in relation to their relevance to the study.

 Discussion: A dedicated discussion section should be added to provide insights into the findings of the study, as well as any limitations or areas for future research. This will help to contextualize the results within existing literature and provide a critical analysis of their implications.

Conclusion: The conclusion section should summarize the key findings of the study, as well as any implications for practice or policy. It should also highlight any limitations or areas for future research, as well as any recommendations for further investigation. Overall, this section should provide a clear and concise overview of the study's main contributions to knowledge in the field.

Author Response

Thank you very much for your comments. In response to your valuable feedback, modifications were made in accordance with your comments:

-The title of the manuscript has been revised according to your feedback: Safety in Public Open Green Spaces in Fortaleza, Brazil: A Data Analysis.

-The introduction has been extensively modified to clearly outline the research gap and objectives. A systematic literature review has been incorporated to demonstrate the novelty and significance of the study, ensuring that the data presented is organized logically and coherently.

-An additional reference has been incorporated into the literature review. We need to address the challenge of sourcing relevant literature on the article's main topic within the context of Latin America. Moreover, multiple references spanning various subjects have been gathered and critically analyzed to enhance the comprehension of the article's primary topic.

-The final paragraph of the introduction has been revised for clarity, now explicitly stating the study's aims and objectives. This revision offers a clear roadmap for the research, aligning with the intended goals of the study.

-The methodology section has been expanded. Additionally, the process for photo selection from social media is now clearly detailed, including the time period, search terms, and any limitations or exclusions.

-The results section has been restructured for logical coherence, with clear headings and subheadings. It directly correlates with the research objectives. The inclusion of newspaper data in the text contextualizes the situation of smartphone robberies in Brazil for an additional reviewer. It is clarified that these are not the author's own results. Still, their presence is vital for providing readers with an understanding of the background scenario, thereby elucidating the rationale behind conducting this research based on photographs from social media.

-A dedicated discussion section has been added, providing insights into the findings, limitations, and potential areas for future research. This section offers a critical analysis and contextualizes the results within the existing literature.

-The conclusion has been improved to summarize the key findings and add the study's main contributions. It also outlines the limitations and recommends areas for further investigation.

Reviewer 3 Report

Comments and Suggestions for Authors

1. Line 177:Jan Gehl (2010) says that……. The reference was incorrectly labeled.

2. Figure 3 needs to supplement the basic elements such as legend, scale, and north pointer.

3. In 3. Results: The position of Figure 4 needs to be adjusted so that it does not appear before annotations in the text.

4. The scales in Figure 4 and Figure 7 are not clear.

5. Judging from the structure of this paper, a Discussion section is missing and needs to be added.

Author Response

Thank you very much for your comments. In response to your valuable feedback we have made the following changes:

-The citation for Jan Gehl (2010) mentioned in Line 177 has been accurately corrected to reflect the appropriate reference, ensuring bibliographic precision.

-Figure 3 has been updated to include a scale and north pointer for better geographical orientation. We opted not to include a legend as the map utilizes a color gradient to represent percentages, which we believe is self-explanatory and aligns with the design of the figure.

- The positioning of Figure 4 within the manuscript has been adjusted to ensure it follows the relevant text annotations. This change enhances the logical flow and visual coherence of the Results section.

-The scales in Figures 4 and 7 have been revised for clarity. These changes ensure that the figures accurately represent the spatial dimensions and data they are intended to convey.

-A Discussion section has been added to the manuscript. This section provides an in-depth analysis of the findings, contextualizes them within the broader research field, and the conclusion addresses potential limitations and avenues for future research.

Reviewer 4 Report

Comments and Suggestions for Authors

The authors should be congratulated on improving their manuscript, and this has become an interesting paper. It is conventional in some contexts for authors to write about the limitations of their work, and I recommend that these authors deal with my criticism of their questionnaire study by acknowledging the limitations of a self-selected population of respondents, with comment on why such a population might be unrepresentative of the background population. For instance, perhaps respondents are more likely to be wealthier than the median, more socially engaged, more able to use leisure time in public parks, etc, etc.

The Instagram study may also have some limitations, in that photographs posted on Instagram may not be representative of all the photographs taken in the park. The very marked concentrations shown in new Figure 6 mitigates against that as a major disadvantage, but again, comments on limitations would be valuable. Are there particular artefacts worthy of photographsy in those areas, such as statuary or gardens. Are the Instagram photographs predominantly portraits or self-portraits, or of still life? It might even be useful (perhaps as a future study, because it could represent a lot of work), to determine whether personal photographs have the same geographic distribution as photographs of still life. Do we know whether the concentration of photographs in a few areas indicate greater security in those areas, rather than greater photogenicity?

Comments on the Quality of English Language

Generally acceptable, with a few idiosyncrasies that can be sorted out by a subeditor.

Author Response

Thank you very much for your comments. In response to your valuable feedback, we recognize the limitations inherent in our questionnaire study, particularly the self-selected nature of the respondent population. Due to geographical constraints, online data collection was our primary method, leading to potential unrepresentativeness in terms of socioeconomic status. We attempted outreach through Facebook communities but received no responses. These limitations, and their implications for the representativeness of our sample, are acknowledged and discussed in the manuscript.

Regarding the nature of why people take pictures, our analysis primarily focuses on the aspect of security in the park. The act of taking out a phone in a public space in Brazil, given the security concerns, implies a level of perceived safety. We correlate this with environmental aspects and the socio-economic context of different areas of the park. This approach helps to understand how environmental quality and safety perceptions vary across the park. Regarding data collection, our reach was limited to the first 150 Instagram posts due to analytical tool constraints, which is a noted limitation. The types of activities depicted in these photos, whether selfies or otherwise, were categorized based on the activity and location, not the style of photography, and can be a bigger study in the future. We acknowledge that specific attractions or photogenic features may skew the distribution of photographed locations. The types of photographs – whether portraits, self-portraits, or still life – and their geographic distribution are noted in the manuscript, although a detailed analysis of this aspect was beyond the scope of our current study.

Round 3

Reviewer 2 Report

Comments and Suggestions for Authors

After reviewing the paper, I can say that the authors have made significant improvements in terms of structure and quality. The paper is now appropriate for publication. However, there are two minor points that the authors should consider:

 1. The authors should compare the outcome of their paper with similar research. This will help to establish the significance and novelty of their findings.

 2. The citation in the conclusion is meaningless as it does not add any value to the discussion. The authors should ensure that the conclusion are relevant and contribute to the overall argument of the paper. Overall, I would like to commend the authors on their efforts

Author Response

We are grateful for your commendations and have taken care to address the two minor points you raised. These final adjustments enhance the robustness and clarity of our study, ensuring that it not only adheres to high academic standards but also meaningfully contributes to the field.

  1. We appreciate your suggestion to compare our findings with similar studies. In response, we have added a comparative analysis in the discussion with research conducted in a geographically distant city, yet yielding similar results.
  2. Regarding the citation in the conclusion, we understand your concern about its relevance. This citation was initially included based on feedback from another reviewer. We have since revised its context and placement to ensure that it contributes meaningfully to the overall argument and cohesively rounds out the conclusion of our paper.

Reviewer 4 Report

Comments and Suggestions for Authors

Lines 411-4: I don't understand this sentence; I suspect line 413 should end with a full stop, not a comma

Line 570: it needs to be acknowledged that this was a convenience sample, and it would be good for the following sentence to indicate that a more formal sampling process might be used in the larger study

Comments on the Quality of English Language

The English is  understandable, but there are a scattering of idiosyncrasies that a sub-editor should be able to deal with.

Author Response

We are grateful for your continued feedback, which has been instrumental in refining our manuscript. These latest revisions address your concerns and further strengthen the paper’s academic quality.

  1. Thank you for pointing out the unclear sentence structure in lines 411-4. We have revised this section for clarity, ensuring that the punctuation accurately reflects the intended meaning.
  2. the sampling method used, as highlighted in line 570, we have explicitly acknowledged that this was a convenience sample. Additionally, we have included a statement indicating the intention to employ a more formal sampling process in future, larger-scale studies.